# Ionotropic Receptor-dependent moist and dry cells control hygrosensation in *Drosophila*

**Zachary A Knecht**[1,2,3], **Ana F Silbering**[4], **Joyner Cruz**[1,2,3], **Ludi Yang**[1,2,3], **Vincent Croset**[4†], **Richard Benton**[4*], **Paul A Garrity**[1,2,3*]

[1]National Center for Behavioral Genomics, Brandeis University, Waltham, United States; [2]Volen Center for Complex Systems, Brandeis University, Waltham, United States; [3]Department of Biology, Brandeis University, Waltham, United States; [4]Center for Integrative Genomics, Faculty of Biology and Medicine, University of Lausanne, Lausanne, Switzerland

**Abstract** Insects use hygrosensation (humidity sensing) to avoid desiccation and, in vectors such as mosquitoes, to locate vertebrate hosts. Sensory neurons activated by either dry or moist air ('dry cells' and 'moist cells') have been described in many insects, but their behavioral roles and the molecular basis of their hygrosensitivity remain unclear. We recently reported that *Drosophila* hygrosensation relies on three Ionotropic Receptors (IRs) required for dry cell function: IR25a, IR93a and IR40a (Knecht et al., 2016). Here, we discover *Drosophila* moist cells and show that they require IR25a and IR93a together with IR68a, a conserved, but orphan IR. Both IR68a- and IR40a-dependent pathways drive hygrosensory behavior: each is important for dry-seeking by hydrated flies and together they underlie moist-seeking by dehydrated flies. These studies reveal that humidity sensing in *Drosophila*, and likely other insects, involves the combined activity of two molecularly related but neuronally distinct hygrosensing systems.

**\*For correspondence:** Richard.
Benton@unil.ch (RB); pgarrity@
brandeis.edu (PAG)

**Present address:** †Centre for
Neural Circuits and Behaviour,
University of Oxford, Oxford,
United Kingdom

**Competing interests:** The
authors declare that no
competing interests exist.

**Reviewing editor:** Mani
Ramaswami, Trinity College
Dublin, Ireland

## Introduction

Hygrosensation is a critical sensory modality for all animals, particularly for insects, whose small bodies and large surface area to volume ratios make dehydration a constant threat. Environmental humidity levels are therefore a key determinant of where and when a given species of insect will thrive (*Chown et al., 2011*). Hygrosensory cues are also fundamental for host-seeking by disease-transmitting mosquitoes (*Brown, 1966*). Despite its importance, the molecular and cellular basis of hygrosensation in insects has remained unclear. In a recent *eLife* paper (*Knecht et al., 2016*)—in parallel with a related study (*Enjin et al., 2016*)—we reported that members of the Ionotropic Receptor (IR) family of variant ionotropic glutamate receptors (*Benton et al., 2009*; *Rytz et al., 2013*) are critical for hygrosensing in *Drosophila melanogaster* (*Knecht et al., 2016*). We identified a set of dry-air-activated neurons ('dry cells') in the sacculus, an internal sensory structure within the antenna, and demonstrated that their capability to sense changes in humidity, as well as the behavioral responses of flies in humidity gradients, require three highly conserved receptors: IR25a, IR93a and IR40a (*Knecht et al., 2016*).

Electrophysiological studies in larger insects suggest that hygrosensation also involves sensory neurons activated by high humidity levels ('moist cells') (*Tichy and Gingl, 2001*). In this study, we identify moist cells in the *D. melanogaster* sacculus and show that their hygrosensitivity requires IR25a and IR93a as well as a previously uncharacterized but evolutionarily conserved receptor, IR68a. We further show that hygrosensory behavior is driven by a combination of IR68a-dependent moist

sensing and IR40a-dependent dry sensing in a manner that varies with the hydration state of the animal.

## Results and discussion

### An *Ir68a* reporter is expressed in candidate moist cells in the *Drosophila* sacculus

To identify cells and receptors involved in moist sensing, we hypothesized that this modality, like dry sensing, involves a conserved IR for which no chemical ligand had been identified. IR68a was an excellent candidate, as this receptor has been conserved across ~350 million years of insect evolution (*Rytz et al., 2013*). Moreover, previous RT-PCR studies (*Croset et al., 2010*) as well as transcriptomic analyses (*Menuz et al., 2014*; *Shiao et al., 2013*) detected *Ir68a* expression in the antenna.

As moist and dry cells are housed in the same sensilla in other species (*Altner and Loftus, 1985*), we anticipated that moist cells in *Drosophila* should be located in the antennal sacculus adjacent to the IR40a-expressing dry cells. While attempts to generate IR68a antisera were unsuccessful, we found that an *Ir68a-Gal4* transgene, containing the putative *Ir68a* promoter, drives expression in a population of neurons that innervate chamber II of the sacculus (*Figure 1a–b*, *Figure 1—figure supplement 1*). Importantly, these *Ir68a-Gal4*-expressing cells are intermingled with, but distinct from *Ir40a*-expressing neurons (*Figure 1c*), consistent with a role as moist cells in the sensilla of chamber II.

IR40a-positive neurons express two co-receptors, IR25a and IR93a (*Knecht et al., 2016*). Immunostaining revealed that IR25a and IR93a proteins are also present in *Ir68a-Gal4*-expressing neurons (*Figure 1d–e*). The cell bodies of sacculus neurons show heterogeneous levels of IR25a and IR93a, with generally lower levels detected in *Ir68a-Gal4*-positive cells; this might reflect differences in the overall IR expression between cells or the efficiency of IR transport to sensory processes.

In the brain, *Ir68a-Gal4*-labeled neurons project to the antennal lobe, terminating in a discrete region near its ventrolateral edge (*Figure 1f*). This region is distinct from the one innervated by IR40a-positive neurons (*Figure 1g–j*) and does not appear to correspond to any previously characterized glomerulus (*Grabe et al., 2015*; *Münch and Galizia, 2016*), consistent with a novel sensory function for these neurons.

### *Ir68a-Gal4* neurons are activated by moist air

To examine the physiological sensitivity of *Ir68a-Gal4*-expressing neurons to humidity changes, we performed calcium imaging in their axon termini using GCaMP6m (*Figure 2a–b*). We observed robust, non-adapting increases in GCaMP fluorescence upon switching from low to high humidity air (7% to 90% relative humidity [RH]), and robust, non-adapting decreases in fluorescence upon switching from high to low humidity air (*Figure 2b–d*). These humidity-dependent calcium responses are opposite of those of IR40a-expressing dry cells (*Enjin et al., 2016*; *Knecht et al., 2016*), indicating that *Ir68a-Gal4* neurons correspond to moist cells.

### Moist cell responses require IR68a, IR25a and IR93a, but not IR40a

To examine the function of IR68a, we obtained *Ir68a* mutants (*Figure 1—figure supplement 1*). Calcium imaging in this mutant background revealed a complete loss of sensitivity of *Ir68a* neurons to humidity changes (*Figure 2c–d*), which was restored by a wild-type *Ir68a* transgene (*Figure 2c–d*, *Figure 1—figure supplement 1*). IR68a is therefore essential for hygrosensory transduction in moist cells.

We previously showed that IR25a and IR93a are required for hygrosensing by dry cells (*Knecht et al., 2016*). Consistent with the expression of these receptors in *Ir68a-Gal4* neurons, moist cell responses were also eliminated in *Ir25a* and *Ir93a* mutants, and these defects could also be restored by the corresponding rescue constructs (*Figure 2c–d*). By contrast, IR40a, which is required for dry cell responses (*Knecht et al., 2016*), was dispensable for the responses of moist cells (*Figure 2c–d*). Similarly, loss of IR68a had no effect on dry cell responses to humidity changes (*Figure 2e–f*). Together these data define two distinct classes of hygrosensory neurons in the sacculus: IR68a-dependent moist cells and IR40a-dependent dry cells.

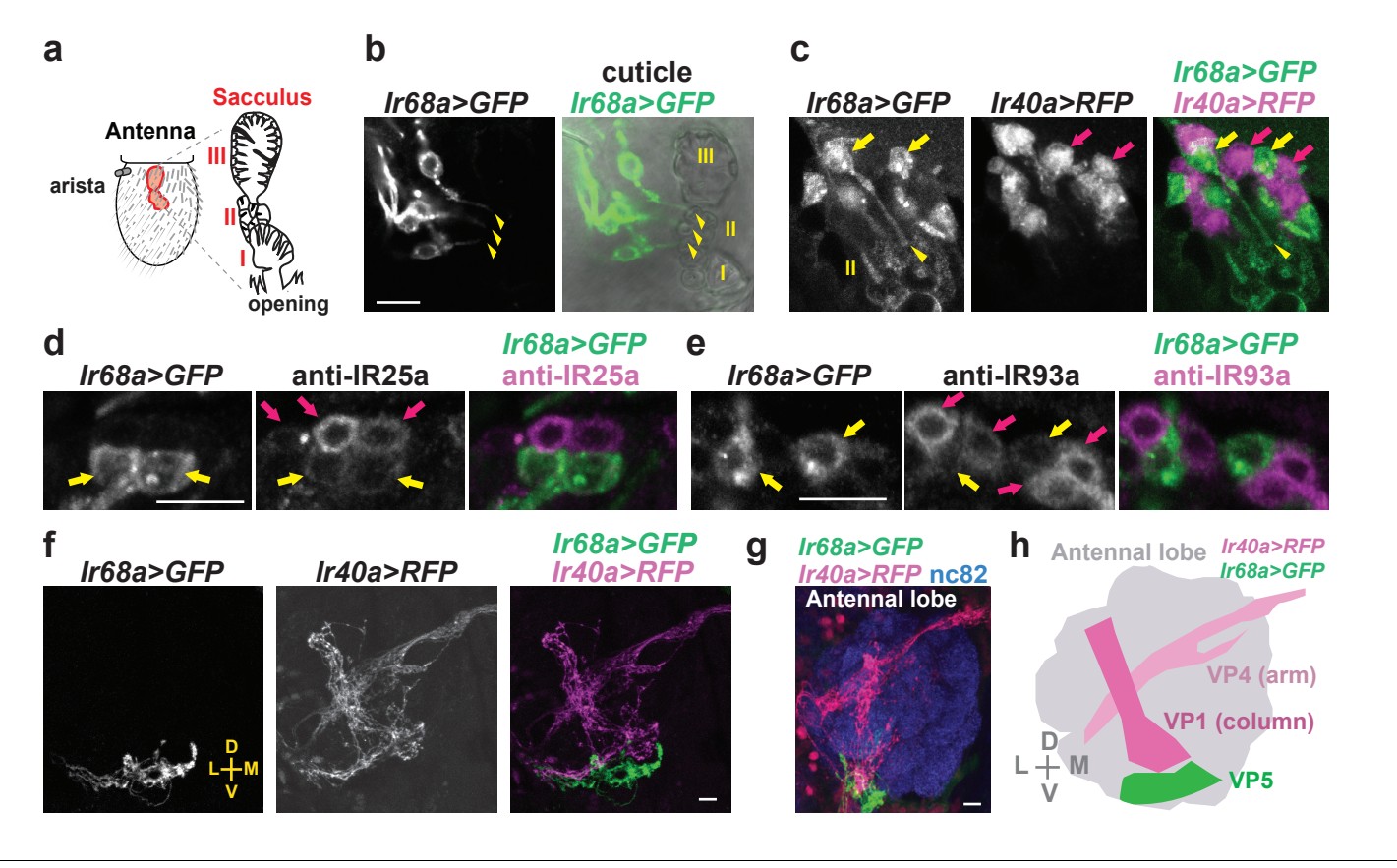

**Figure 1.** *Ir68a* and *Ir40a* reporters are expressed by neighboring neurons in the *Drosophila* sacculus. (a) Left: schematic of the adult *Drosophila* antenna, showing the location of the sacculus (red) inside the antenna. Right: the sacculus contains three chambers (I, II, III) lined with sensilla of various morphologies (modified from *Shanbhag et al. [1995]*) and *Knecht et al. [2016]*). (b) Immunostaining of the antenna of a *Ir68a-Gal4/UAS-myr:GFP* (*Ir68a>GFP*) fly (left). *Ir68a* is expressed in neurons that send processes to sacculus chamber II (9.8 ± 0.4 neurons (mean ± SEM); n = 8 antennae). Arrowheads denote sensory endings. Scale bar in all panels indicates 10 μm. Cuticle autofluorescence outlines sacculus chambers in the right panel. (c) Immunostaining of *LexAop-RFP(II);Ir68a-Gal4/Ir40a-LexA,UAS-myr:GFP* (*Ir68a>GFP, Ir40a>RFP*) flies reveals non-overlapping expression in cells adjacent to chamber II. Arrows indicate select cell bodies. Arrowhead marks an *Ir68a>GFP*-labeled dendrite projecting into a chamber II sensillum. Cuticle autofluorescence outlines sensilla of the sacculus in the GFP channel. (d,e) Immunofluorescence on antennal cryosections reveals overlapping expression of *Ir68a>GFP* with IR25a protein (d) and IR93a protein (e) in sacculus neurons. Yellow arrows denote cells that detectably co-express *Ir68a>GFP* and IR25a (d) or *Ir68a>GFP* and IR93a (e). Purple arrows denote cells expressing only IR25a or IR93a, reflecting their broader expression in the sacculus, including co-expression with IR40a (*Knecht et al., 2016*). (f–g) *Ir68a>GFP*-labeled and *Ir40a>RFP*-labeled axons project to distinct regions of the antennal lobe. In panel g, the antennal lobe neuropil is labeled using nc82. (h) Schematic indicating the relative position of *Ir40a* neuron and *Ir68a* neuron projections; VP1 and VP4 glomerular nomenclature is from *Grabe et al. (2015)*. The *Ir68a>GFP*-labeled glomerulus (which has not been previously noted) has been denoted VP5 to maintain consistency with existing nomenclature. D-dorsal; V-ventral; L-lateral; M-medial. The organization of the *Ir68a* locus including the region used to generate *Ir68a-Gal4* is provided in *Figure 1—figure supplement 1*.

The following figure supplement is available for figure 1:

**Figure supplement 1.** Organization of *Ir68a* locus.

## IR68a is required for hygrosensory behavior

The role of IR68a in behavioral responses to humidity differences was assessed by quantifying the distribution of flies in a humidity gradient (~67% to ~96% RH) (*Figure 3a*) (*Knecht et al., 2016*). Consistent with our previous report (*Knecht et al., 2016*), wild-type flies exhibit robust dry-seeking behavior, and loss-of-function mutations in *Ir25a*, *Ir93a* and *Ir40a* significantly reduce this response (*Figure 3b*). Loss-of-function mutations in *Ir68a* caused a similar decrease in dry preference, a defect rescued by the introduction of a wild-type *Ir68a* transgene (*Figure 3b*). Thus, IR68a is critical for

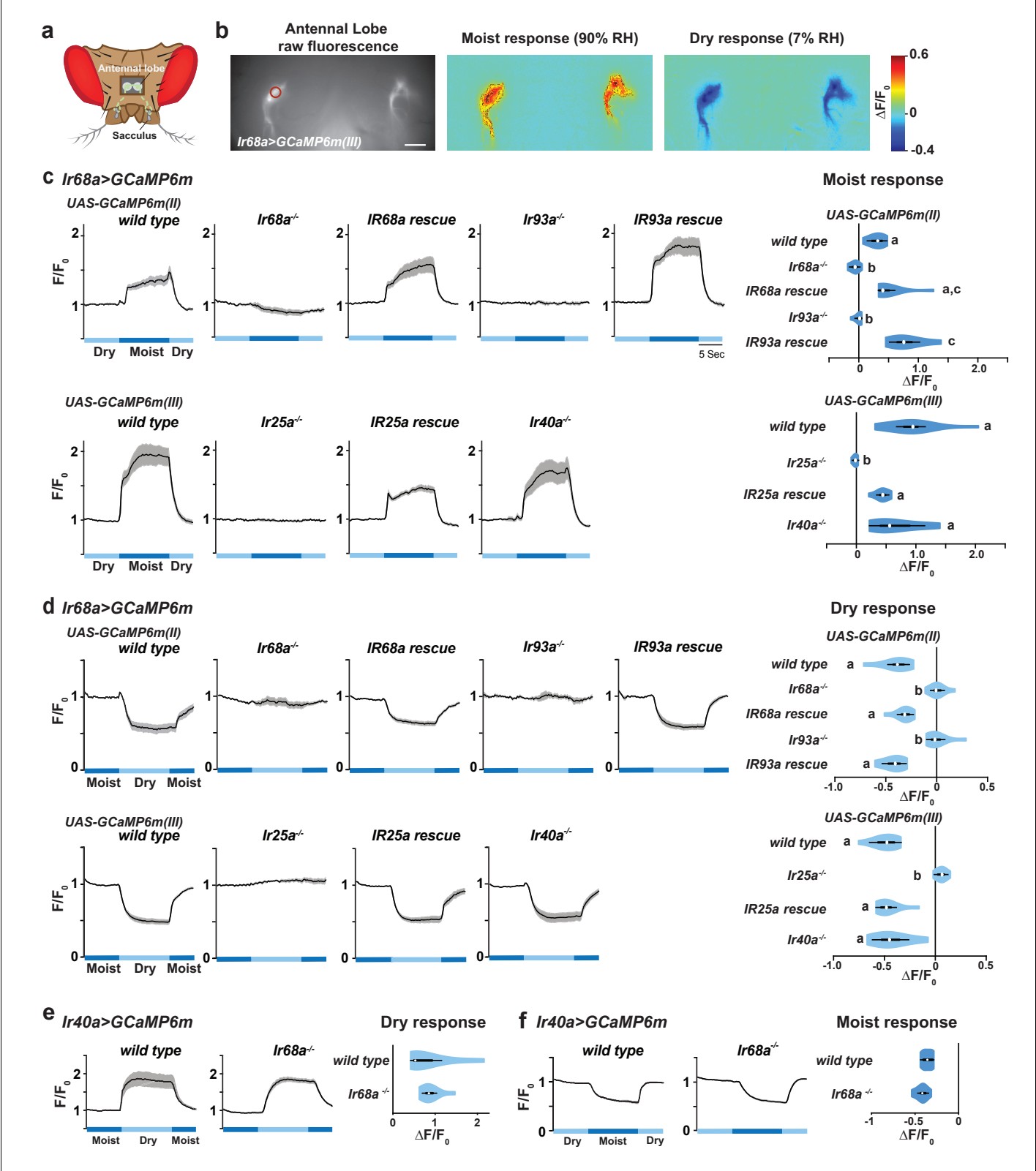

**Figure 2.** *Ir68a* is required for humidity detection by moist cells. (**a**) Schematic of the *Drosophila* head (viewed from above) illustrating the projection of *Ir68a-Gal4*-labeled neurons (green) from the sacculus to the antennal lobes in the brain, visualized through a hole cut in the head cuticle. (**b**) Left panel: Raw fluorescence image of *Ir68a*-labeled axons (in an *Ir68a-Gal4;UAS-GCaMP6m(III)* animal) innervating the antennal lobe. The circle indicates the position of the ROI used for quantification. Middle and right panels: color-coded images (reflecting GCaMP6m relative fluorescence intensity changes)
*Figure 2 continued on next page*

*Figure 2 continued*

of responses to a switch from 7% to 90% RH ('Moist response') and to a switch from 90% to 7% RH ('Dry response'). Scale bar is 10 μm. (c,d) Moist-elicited (c) and dry-elicited (d) fluorescence changes in the region of interest in panel (b) (moist = 90% RH, dry = 7% RH). Left panels: Traces represent mean ± SEM. Right panels: Quantification of responses. Letters denote statistically distinct groups (p≤0.05) Steel-Dwass. Data obtained using *UAS-GCaMP6m(II)* and *UAS-GCaMP6m(III)* were analyzed separately. Moist-responses were calculated as [F/$F_0$ at 90% RH (average F/$F_0$ from 4.5 to 6.5 s after shift to 90% RH)] - [F/$F_0$ at 7% RH (average F/$F_0$ from 3.5 to 1 s prior to shift to 90% RH)]. Dry-responses were quantified using the converse calculation. Violin plots: internal white circles show median; black boxes denote 25th to 75th percentiles; whiskers extend 1.5x the interquartile range. Genotypes: *wild type (UAS-GCaMP6m(II))*: Ir68a-Gal4,UAS-GCaMP6m(II) (n = 7 animals). *Ir68a$^{-/-}$*: UAS-GCaMP6m(II);Ir68a-Gal4,Ir68a$^{c04139}$/Ir68a$^{c04139}$ (n = 9). *Ir68a rescue*: Ir68a$^+$ rescue transgene(II)/UAS-GCaMP6m(II);Ir68a-Gal4,Ir68a$^{c04139}$/Ir68a$^{c04139}$ (n = 8). *Ir93a$^{-/-}$*: UAS-GCaMP6m(II);Ir68a-Gal4, Ir93a$^{MI05555}$/Ir93a$^{MI05555}$ (n = 8). *Ir93a rescue*: UAS-GCaMP6m(II);Ir68a-Gal4,Ir93a$^{MI05555}$/UAS-mCherry:Ir93a,Ir93a$^{MI05555}$) (n = 6). *wild type (UAS-GCaMP6m(III))*: Ir68a-Gal4,UAS-GCaMP6m(III) (n = 10 animals). *Ir25a$^{-/-}$*: Ir25a$^2$;Ir68a-Gal4,UAS-GCaMP6m(III) (n = 8). *Ir25a rescue*: Ir25a$^2$,Ir25aBAC/Ir25a$^2$; Ir68a-Gal4,UAS-GCaMP6m(III) (n = 8). *Ir40$^{-/-}$*: Ir40a$^1$;Ir68a-Gal4,UAS-GCaMP6m(III) (n = 8). (e,f) Moist-elicited (e) and dry-elicited (f) fluorescence changes in Ir40a-Gal4-labeled dry receptors, as in panels c-d. Genotypes: *wild type*: Ir40a-Gal4/UAS-GCaMP6m (n = 9). *Ir68a$^{-/-}$*: Ir40aGal4/UAS-GCaMP6m; Ir68a$^{c04139}$ (n = 9). Ir68a mutant alleles and genomic rescue fragment are shown in **Figure 1—figure supplement 1**. Source data for summary graphs are provided in **Figure 2—source data 1**.

The following source data is available for figure 2:

**Source data 1.** Calcium imaging results.

behavioral responses to humidity. Notably, because IR40a-dependent dry cell responses persist in *Ir68a* mutants (**Figure 2e–f**), dry-sensing neurons appear to be insufficient to guide hygrotaxis, at least in this assay. Similarly, moist cell function alone is insufficient to support normal hygrotaxis, because *Ir40a* mutants display hygrosensory behavioral impairment (**Figure 3b**), despite having physiologically-active moist cells (**Figure 2c–d**). Flies lacking both *Ir68a* and *Ir40a* displayed defects similar to the single mutants (as well as *Ir25a* or *Ir93a* mutants in which both moist and dry pathways are eliminated) (**Figure 3b**). Together these results indicate that intact IR40a- and IR68a-dependent pathways are required for wild-type hygrotaxis.

## Hydration state alters the impact of IR-dependent moist and dry sensing on behavior

The behavioral requirement for both IR40a and IR68a raised the question of whether their combined activity was an obligate feature of the hygrosensory system or whether the function of a single pathway suffices under some conditions. Hydration state dramatically alters insect responses to humidity (**Chown et al., 2011**), and *Drosophila* prefer significantly moister environments when dehydrated (**Perttunen and Erkkila, 1952**). Consistent with those observations, animals previously subjected to desiccation stress became strongly moist-seeking in our assay (reflected in the shift of Dry Preference to negative values) (**Figure 4a–b**). Null mutations in either *Ir25a* or *Ir93a* (which disrupt both dry and moist cell functions) abolished this moist preference, indicating that dehydrated flies still rely on IR-dependent hygrosensing (**Figure 4b**). By contrast, moist-seeking behavior persisted in dehydrated *Ir40a* and *Ir68a* single mutants, although it was slightly reduced in *Ir40a* mutants (**Figure 4b**). However, in *Ir40a;Ir68a* double mutant animals, moist-seeking was completely abolished (**Figure 4b**). Thus, while hydrated flies are dependent on both *Ir40a*- and *Ir68a*-dependent signaling, flies experiencing desiccation stress exhibit significant moist-seeking as long as one pathway is operative.

## Conclusions

Our data establish a central role for an evolutionarily conserved set of IRs in physiological and behavioral responses to humidity in *Drosophila*. IR25a and IR93a are important for humidity detection by both moist and dry cells, while IR68a and IR40a are specifically required for moist or dry cell function, respectively. Together, these receptors are essential for driving hygrotaxis, as the loss of either IR25a or IR93a, or the combined loss of IR40a and IR68a, eliminates responses to humidity in our assays. This work reinforces a model in which the broadly expressed co-receptors IR25a and IR93a act with more selectively expressed IRs that determine the specificity for different chemo-, thermo-, or hygrosensory cues (**Abuin et al., 2011**; **Knecht et al., 2016**; **Ni et al., 2016**). Attempts to replace

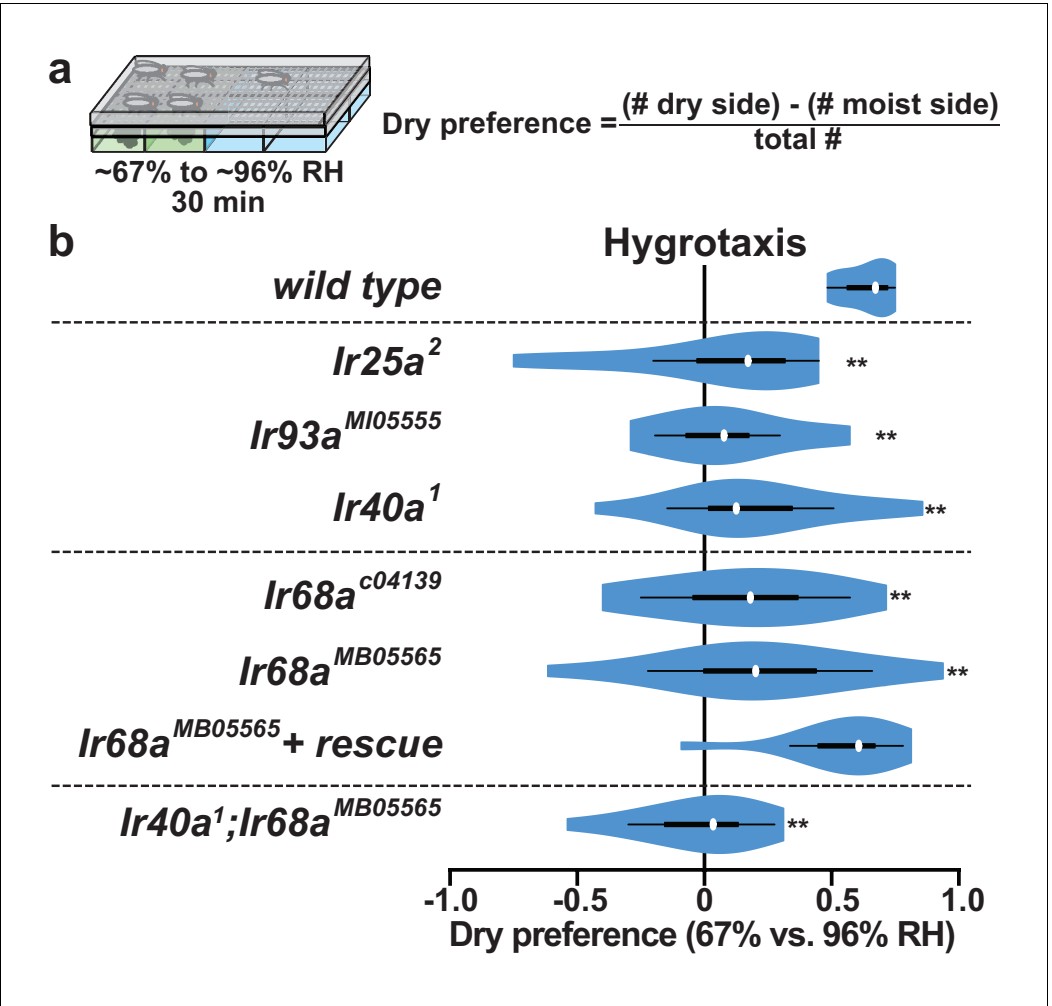

**Figure 3.** *Ir68a* is required for hygrosensory behavior. (**a**) Schematic of hygrotaxis assay.~67% to ~96% RH gradients were generated as described (*Knecht et al., 2016*). Dry preference was quantified by counting flies on either side of chamber midline. 25–35 flies were used per assay. (**b**) Dry preference of hydrated flies. Asterisks denote values that are statistically distinct from *wild type* (**p<0.01, Steel with control). *wild type* (n = 12 assays). *Ir25a²* (n = 12). *Ir93a^{MI05555}* (n = 11). *Ir40a¹* (n = 15). *Ir68a^{c04139}* (n = 16). *Ir68a^{MB05565}* (n = 14). *Ir68a^{MB05565} + rescue* (*Ir68a⁺ rescue transgene(II);Ir68a^{MB05565}*) (n = 12). *Ir40a¹;Ir68a^{MB05565}* (n = 15). Source data for summary graph are provided in *Figure 3—source data 1*.

The following source data is available for figure 3:

**Source data 1.** Hygrotaxis behavior data.

IR68a with IR40a in moist cells or IR40a with IR68a in dry cells (by removing the endogenous IR through mutation and misexpressing the other IR in its place) yielded neurons that did not respond to either moist or dry air (ZK and PAG, unpublished data). These failures may reflect the absence of additional moist/dry cell-specific co-factors or structures that are crucial for IR-dependent hygrosensing. Further mechanistic insights into humidity detection by IRs will require reconstitution of IR-dependent hygrosensory responses in expression systems that permit structure/function analyses.

It is notable that the same IRs mediate hygrotaxis regardless of the hydration state of the fly, even though dehydration switches the valence of behavioral responses in a humidity gradient. These observations suggest that the moist and dry hygrosensory pathways do not simply promote attraction or aversion when activated. We propose that the information conveyed by these peripheral neurons is combined in the brain with signals from internal hydration sensors to determine how the

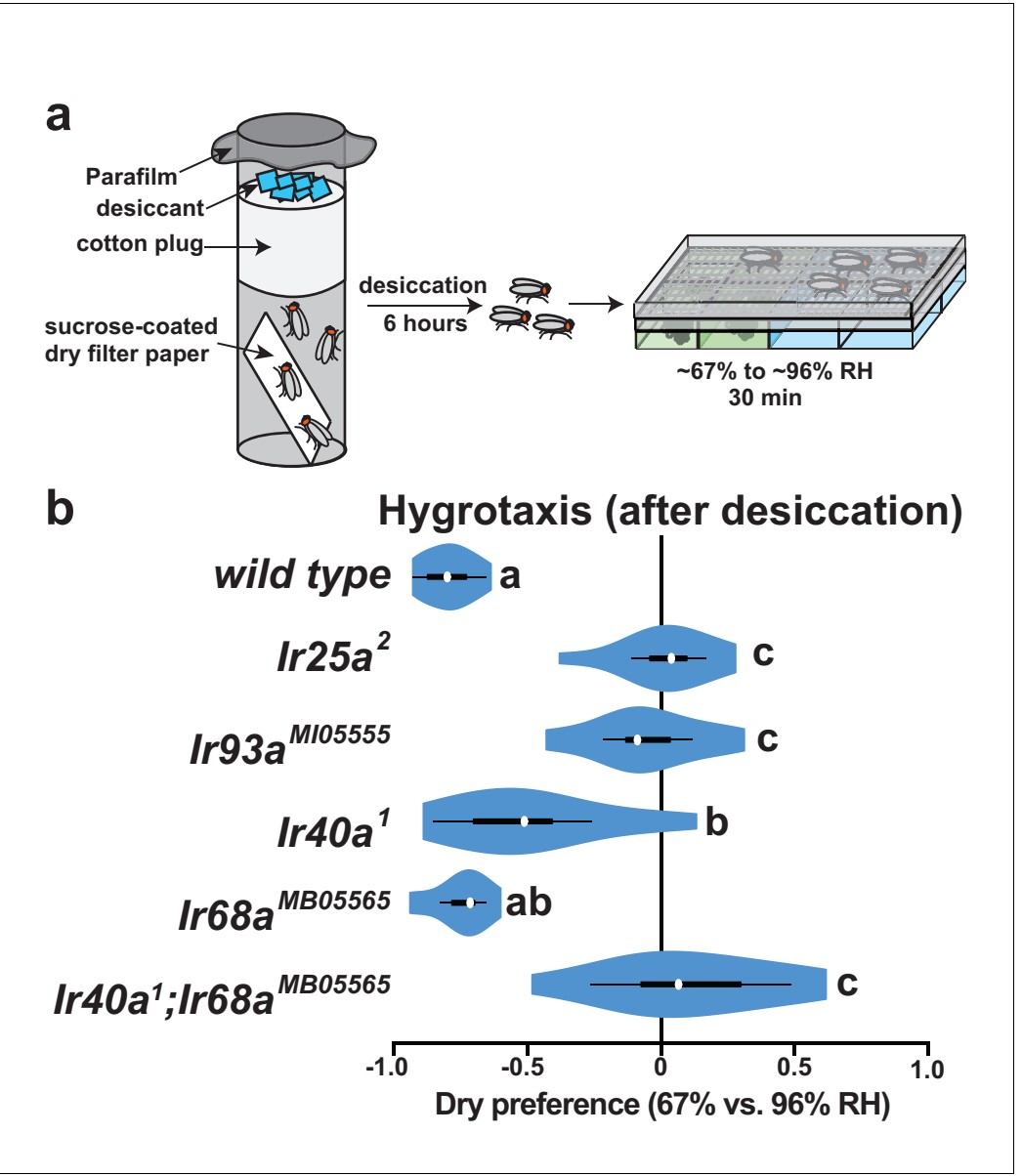

**Figure 4.** *Ir40a* and *Ir68a* each contribute to hygrotaxis in dehydrated flies. (a) Schematic of hygrotaxis assay using dehydrated flies. (b) Hygrotaxis behavior in dehydrated flies. Letters denote statistically distinct groups (p<0.01, Tukey HSD). Genotypes: *wild type* (n = 15 assays). *Ir25a²* (n = 13). *Ir93a^MI05555* (n = 14). *Ir40a¹* (n = 12). *Ir68a^MB05565* (n = 11). *Ir40a¹;Ir68a^MB05565* (n = 12). Source data for summary graph are provided in *Figure 4—source data 1*.
The following source data is available for figure 4:

**Source data 1.** Hygrotaxis behavior data.

animal responds to moisture. Internal osmolarity-sensing neurons that influence water consumption have recently been identified in *Drosophila* (*Jourjine et al., 2016*), and similar classes of internal sensors of water balance seem likely to be involved in setting the preference for moist versus dry environments. Future central mapping of the pathways that influence water-seeking behaviors will begin to reveal how the animal monitors and adjusts its hydration state to maintain optimal concentrations of this essential biological solvent.

## Material and methods

### *Drosophila* strains

*Ir25a$^2$* (*Benton et al., 2009*) (RRID:BDSC_41737), *Ir25a-BAC* (*Chen et al., 2015*), *Ir40a-Gal4* (*Silbering et al., 2011*) (RRID:BDSC_41737), *Ir40a-LexA* (*Silbering et al., 2016*), *Ir40a$^1$* (*Silbering et al., 2016*), *Ir93a$^{MI05555}$* (*Knecht et al., 2016*) (RRID:BDSC_42090), *UAS-Ir93a* (*Knecht et al., 2016*), *UAS-GCaMP6m(II)* (*P[20XUAS-IVS-GCaMP6m]attP40*) (RRID:BDSC_42748), *UAS-GCaMP6m(III)* (*P[20XUAS-IVS-GCaMP6m]VK00005*) (RRID:BDSC_42750) (*Chen et al., 2013*)), and *UAS-myr:GFP* (*P[10UAS-IVS-myr::GFP]attP1*) (*Pfeiffer et al., 2012*) were previously described. *LexAop-RFP* (P[lexA-2xmRFP.nls]2) (RRID:BDSC_29956) and *Ir68a$^{MB05565}$* (RRID:BDSC_26031) were obtained from the Bloomington *Drosophila* Stock Center, and *Ir68a$^{c04139}$* was obtained from the Exelixis Collection at Harvard Medical School.

### Molecular biology

*Ir68a-Gal4* was generated by PCR amplification of a 1040 bp genomic sequence directly upstream of the *Ir68a* translation start site (using 5'-cggccgcCACGTCGTCGTCCGCATTAC and 5'-gcggccgcCCTTTCGCCGCCAAACGCAA), which was cloned into pGEM-T Easy (Promega, Madison, WI), sequenced, and sub-cloned as a *NotI* fragment into *pGal4 attB* (*Croset et al., 2010*); this construct was integrated into attP2 (*Markstein et al., 2008*) (RRID:BDSC_8622). The *Ir68a$^+$* rescue transgene contains *Ir68a* genomic sequence from −1040 bp to +4751 bp (+1 denotes *Ir68a* translation start site), which was PCR amplified using 5'-cgttacacgcatgcCACGTCGTCGTCCGCATTACAATATC and 5'- acggaccactctagaTGAAGTGTGGGTGTTTCTCCAACCA. This PCR product was digested with *SbhI* and *XbaI*, and used to replace the UAS-hsp70 promoter sequences of *pUAST-attB*, which had been excised by *SbhI* and *XbaI* digest. This *Ir68a$^+$*-attB construct was integrated into attP40 (*Markstein et al., 2008*).

### Immunohistochemistry

Whole mount antennal stainings (*Figure 1b*) were performed as previously described (*Knecht et al., 2016*). Immunostaining of antennal cryosections (*Figure 1c–e*) was performed largely as described (*Saina and Benton, 2013*). Male and female 1- to 3-week-old flies were mounted in OCT (Sakura #4583), 12 µm or 16 µm frozen sections were cut and then fixed in 4% paraformaldehyde at room temperature for 7–10 min. For brain stainings (*Figure 1f–i*), female flies were dissected in 1XPBS before either ~2 min of fixation and immediate mounting, or 5 min of fixation and staining in primary antibody. All samples were mounted in Vectashield (Vector Labs) for confocal microscopy (Leica SP5 or Zeiss LSM710). The following antibodies were used: *Figure 1b*: chicken anti-GFP (1:1000, Abcam 13970) and goat anti-Chicken Alexa488 (1:1000, Abcam 150169); *Figure 1e*: mouse anti-GFP (1:1000, Invitrogen A11120), rabbit anti-IR93a (1:3000) (*Knecht et al., 2016*)), goat anti-mouse Alexa488 (1:1000, A11029 Invitrogen), goat anti-rabbit Cy3 (1:1000, 111-165-144 0 Milan Analytica); *Figure 1c,d,f and g*: chicken anti-GFP (1:200 brains or 1:1000 cryosections, Aves Labs GFP-1020), rabbit anti-DsRed (1:200 brains or 1:1000 cryosections, Clontech #632496), rabbit anti-IR25a (1:100, (*Benton et al., 2009*)), goat anti-chicken Alexa488 (1:200, Life Technologies A-11039), goat anti-rabbit Alexa594 (1:200, Life Technologies A-11037), mouse anti-nc82 (1:500, Developmental Studies Hybridoma Bank) and goat anti-mouse Cy5 (1:200, Jackson Labs #115-176-0030).

### Calcium imaging

Calcium imaging was performed as described (*Knecht et al., 2016*). Data were processed largely as described (*Knecht et al., 2016*) but using a different custom Matlab script (source code file provided as *Source code 1*- Calcium Imaging). For analysis, the average pixel intensity of a nearby background region (BGR) was subtracted from the average pixel intensity of a polygon drawn around the labeled glomerulus (ROI). The first 20 frames (5 s) were used to define baseline fluorescence ($F_0$). $F/F_0$ was calculated using $F_i/F_0$ (frame $i$) = ($ROI_i$ − $BGR_i$)/$F_0$. As quantified imaging data did not conform to normal distribution (assessed by Shapiro-Wilk test, p<0.01), statistical comparisons were performed by Steel-Dwass test using JMP11 (SAS). Samples unresponsive to humidity change were subsequently depolarized by bath application of 0.7 mM KCl to confirm their physiological integrity; animals unresponsive to this positive control were excluded from analysis. Animals were also

excluded if movement artifacts could not be corrected using Stackreg in ImageJ (*Schneider et al., 2012*).

## Behavior

Hygrosensory behavior was assayed as previously described (*Knecht et al., 2016*). Desiccation prior to analysis was performed using a modification of (*Lin et al., 2014*). Flies were sorted and placed in tubes containing a strip of filter paper soaked in 3% sucrose and let to dry. The vial stopper was pushed down below the vial lip, ~0.5 g Drierite spread over it, and Parafilm was placed over the top to seal the vial. Vials were kept in an incubator (25°C, 70% RH) for 6 hr before hygrosensory behavior was assessed as described above. Humidity preference data in *Figure 3* did not conform to normal distributions (assessed by Shapiro-Wilk test, $p < 0.01$) and were analyzed by Steel with control test using JMP11 (SAS). Humidity preference data in *Figure 4* did conform to normal distributions and were analyzed using Tukey HSD using JMP11 (SAS).

## Acknowledgements

We acknowledge the Bloomington *Drosophila* Stock Center (NIH P40OD018537) and the Developmental Studies Hybridoma Bank (NICHD of the NIH, University of Iowa) for reagents. We thank Liliane Abuin for assistance with transgene generation and characterization, Timothy Wiggin and Brian Carey for assistance with Matlab script, and Willem Laursen, Lina Ni, Joyce Rigal, and Lena van Giesen for comments on the manuscript. This work was supported by a grant from the National Institute on Deafness and Other Communication Disorders (F31 DC015155) to ZAK, a Boehringer Ingelheim Foundation Fellowship to VC, European Research Council Starting Independent Researcher and Consolidator Grants (205202 and 615094) and a Swiss National Science Foundation Project Grant (31003A_140869) to RB, the National Institute of General Medical Sciences (P01 GM103770) and the National Institute of Allergy and Infectious Diseases (R01 AI122802) to PAG.

## Additional information

### Funding

| Funder | Grant reference number | Author |
|---|---|---|
| National Institute on Deafness and Other Communication Disorders | F31 DC015155 | Zachary A Knecht |
| Boehringer Ingelheim Stiftung | | Vincent Croset |
| European Research Council | 205202 | Richard Benton |
| European Research Council | 615094 | Richard Benton |
| Schweizerischer Nationalfonds zur Förderung der Wissenschaftlichen Forschung | 31003A_140869 | Richard Benton |
| National Institute of General Medical Sciences | P01 GM103770 | Paul A Garrity |
| National Institute of Allergy and Infectious Diseases | R01 AI22802 | Paul A Garrity |

The funders had no role in study design, data collection and interpretation, or the decision to submit the work for publication.

### Author contributions

ZAK, Conceptualization, Formal analysis, Funding acquisition, Investigation, Visualization, Methodology, Writing—original draft, Writing—review and editing; AFS, Conceptualization, Formal analysis, Investigation, Visualization, Methodology, Writing—original draft, Writing—review and editing; JC, LY, VC, Investigation, Writing—review and editing; RB, Conceptualization, Supervision, Funding acquisition, Investigation, Visualization, Writing—original draft, Writing—review and editing; PAG,

Conceptualization, Formal analysis, Supervision, Funding acquisition, Investigation, Visualization, Writing—original draft, Writing—review and editing

**Author ORCIDs**

Paul A Garrity, http://orcid.org/0000-0002-8274-6564

## Additional files

**Supplementary files**

• Source code 1. Code for Calcium Imaging analysis

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
