## [Decision Letter]

Thank you for submitting your article "Ionotropic Receptor-dependent moist and dry cells control hygrosensation in *Drosophila* for consideration by *eLife*. Your article has been favorably evaluated by K VijayRaghavan (Senior Editor) and two reviewers, one of whom, Mani Ramaswami (Reviewer #1), is a member of our Board of Reviewing Editors. The following individual involved in review of your submission has agreed to reveal their identity: Hubert Amrein (Reviewer #2).

The reviewers have discussed the reviews with one another and the Reviewing Editor has drafted this decision to help you prepare a revised submission.

Summary:

The Research Advance by Knecht and colleagues does an excellent job of establishing new features of hygrosensation mechanisms in *Drosophila*, originally described in the "parent" paper on which this advance is based. This work identifies an IR based hygrosensor expressed in moist cells of the *Drosophila* antennal sacculus. It is a nice, complementary study of the work form the same lab, in which an IR based "dry receptor" was reported, in cells intermingled with the moist cells. However, dryness and wetness sensing neurons differ in the third IR that they express and use for their distinctive sensory purposes. The authors find that the dry and moist receptor share two subunits (Ir25a and IR93a), while a specific IR component mediates either dry (IR40a) or moist (IR68a) air. Both dry and wet sensing neurons together mediate humidity-seeking behaviour.

The above are established with appropriate genetic, behavioural and physiological (GCaMP and ArcLight) studies. An additional observation is that the valence of humid environments is altered depending on the state of dehydration of the animal, a nice observation that sets the stage for analysis of neuromodulatory mechanisms that respond to thirst and mediate thirst-driven behaviours.

Suggested revisions:

1) An important and relevant question is whether dry and wet receptors differ only in their third subunit, or if there are other distinctive subunits as well. This question could be addressed by testing whether expression of IR68a in dry neurons of IR40a mutants can "switch" the response of these neurons from "dry" to "moist", and vice versa, whether expression of IR40a in moist neurons of IR68a mutants can "switch" their response from "moist" to "dry". Has this been done? Why not? If the tools and reagents exist, then can this experiment be done within a reasonable time frame?

---

## [Author Response]

*Suggested revisions:*

*1) An important and relevant question is whether dry and wet receptors differ only in their third subunit, or if there are other distinctive subunits as well. This question could be addressed by testing whether expression of IR68a in dry neurons of IR40a mutants can "switch" the response of these neurons from "dry" to "moist", and vice versa, whether expression of IR40a in moist neurons of IR68a mutants can "switch" their response from "moist" to "dry". Has this been done? Why not? If the tools and reagents exist, then can this experiment be done within a reasonable time frame?*

We performed both of these experiments but they yielded negative results. When we replaced IR68a with IR40a or IR40a with IR68a (i.e., endogenous receptor was removed through mutation, new receptor misexpressed in its place) we saw no responses to humidity changes whatsoever: we could neither rescue nor invert the response of the dry or moist cells. There are many possible reasons for negative ectopic expression data including the absence of cell-specific co-factors or cellular structures (as we discussed in our previous *eLife* paper), but we do not currently have sufficient information on humidity transduction mechanisms to favor a specific explanation.

Interestingly, when we ectopically expressed IR68a in IR40a-expressing dry cells (using Ir40a-Gal4 in an otherwise wild-type animal), these IR68a/IR40a co-expressing cells behaved like a hybrid between “moist cells” and “dry cells”. They were transiently activated by exposure to moist air (e.g., GCaMP fluorescence rose for ~1 sec), but then (as the moist air flow continued) they became inhibited (GCaMP fluorescence dropped below baseline). These cells therefore act partly like “moist cells” and partly like “dry cells”. Similarly, moist cells forced to express IR40a (using Ir68a-Gal4) were transiently activated by dry air (for ~1 sec) (like a “dry cell”) before exhibiting inhibition (like a “moist cell”). Thus, although misexpressing IR40a or IR68a did alter humidity-dependent responses in these experiments, the effects were complex, and we feel it would be prudent to investigate the precise origins of these phenotypes in future work before presenting them in a publication.